# DATASET DISTILLATION FOR MEMORIZED DATA: SOFT LABELS CAN LEAK HELD-OUT TEACHER KNOWLEDGE

**Freya Behrens & Lenka Zdeborová**
Statistical Physics of Computation Laboratory
École polytechnique fédérale de Lausanne (EPFL), Switzerland
`freya.behrens@epfl.ch`

## ABSTRACT

Dataset distillation aims to compress training data into fewer examples via a teacher, from which a student can learn effectively. While its success is often attributed to structure in the data, modern neural networks also memorize specific facts, but if and how such memorized information can be transferred in distillation settings remains less understood. While this transfer may be desirable in some applications, it also raises privacy concerns, where preventing such leakage is crucial. In this work, we show that students trained on soft labels from teachers can indeed achieve non-trivial accuracy on held-out memorized data they never directly observed. This effect persists on structured data when the teacher has not generalized. To understand this effect in isolation, we consider finite random i.i.d. datasets where generalization is a priori impossible and a successful teacher fit implies pure memorization. Still, students can learn non-trivial information about the held-out data, in some cases up to perfect accuracy. For multinomial logistic classification and single layer MLPs, we show this corresponds to the setting where the teacher can be recovered functionally – the student matches the teacher's predictions on all possible inputs, including the held-out memorized data. We empirically show that these phenomena strongly depend on the sample complexity and the temperature with which the logits are smoothed, but persist across varying network capacities, architectures and dataset compositions.

## 1 INTRODUCTION

With the advent of foundation models, it has become of great interest to exploit and transfer their capabilities to other models and finetuning and distillation make this possible in practice. In distillation, a student model is trained on data derived from a teacher model (Hinton et al., 2015; Xu et al., 2024); dataset distillation specifically focuses on finding a minimal training set that achieves high performance for similarly sized teacher and student models (Cazenavette et al., 2023; Yu et al., 2024; Yang et al., 2024). A central mechanism is the use of soft labels, where the teacher's logits are transformed into probability distributions that the student is trained to match (Buciluǎ et al., 2006; Ba & Caruana, 2014; Hinton et al., 2015). This simple idea has been remarkably effective and remains competitive for modern architectures (Gou et al., 2021a; Yu et al., 2024; Xu et al., 2024; Qin et al., 2024). While there have been theoretical attempts to explain the benefits of soft labels (Phuong & Lampert, 2019; Saglietti & Zdeborova, 2022; Menon et al., 2021; Boix-Adsera, 2024; Dissanayake et al., 2025), it remains unclear what exactly the "dark knowledge" (Hinton et al., 2015) in soft labels is, and how to quantify it.

Among the hypotheses on the regularizing benefits of soft labels (Müller et al., 2019; Yuan et al., 2020; Zhou et al., 2021), one line of reasoning suggests that they are effective because they encode latent structure in the data distribution (Phuong & Lampert, 2019; Menon et al., 2021). This view explains empirical successes in image classification and language, where soft labels regularize the student by exposing correlations in the teacher's predictions (Qin et al., 2024; Xu et al., 2024). However, success in large-scale models does not rely on structure alone: such models not only generalize from data distributions but also memorize singular facts and associations (Chen et al., 2024). This raises the question of whether soft labels also convey memorized information, and if students can inherit it during distillation. So far, this issue has been studied mainly from a privacy perspective: soft labels and related outputs may leak memorized training information, enabling the recovery of private

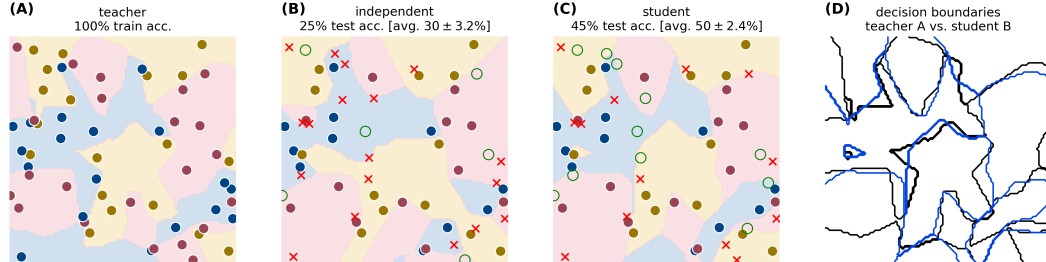

Figure 1: **Information leakage via soft labels.** We examine fully connected networks with ReLU activations and $p = 100$ hidden neurons and biases. A teacher network is trained on 2D input data $\mathcal{D}_{\star}^{\mathrm{T}}$ with i.i.d. random uniform labels drawn from $\{1, 2, 3\}$. **(A)** visualizes $\mathcal{D}_{\star}^{\mathrm{T}}$ and teacher decision boundaries which achieve 100% accuracy. Then, $\mathcal{D}_{\star}^{\mathrm{T}}$ is partitioned into two disjoint sets $\mathcal{D}_{\mathrm{train}}^{\mathrm{S}}$ and $\mathcal{D}_{\mathrm{test}}^{\mathrm{S}}$ $(60\%, 40\%)$. We examine 2 settings: Training student networks via cross-entropy **(B)** on the class label only, making the student independent from the teacher, and **(C)** on soft labels obtained from the teacher via softmax on the logits. While the independent model only achieves trivial accuracy of $\sim 30\%$, students that fit the teacher's soft labels achieve *non-trivial test accuracy* of $\sim 50\%$. Markers indicate data from the test set, and whether it was classified wrongly ($\times$) or correctly ($\circ$). We report averages and the standard error on the mean over 5 runs. **(D)** The decision boundaries for teacher (black) and student (blue). Appendix A contains further examples.

attributes (Ma et al., 2024; Cloud et al., 2025). Yet, these works focus on attack scenarios, and the question of how memorization behaves in benign distillation is still open.
We therefore ask:

> *Do the teacher's soft labels encode memorized knowledge?*
> *– And if yes, can students pick up this non-trivial information?*

To address this, we study memorization transfer in small empirical models that allow for precise control and measurement, rather than attempting to analyze large-scale systems directly. We isolate the role of memorization in distillation with soft labels by training teacher networks to memorize a finite training dataset of input–label pairs. We then distill their "memorized" knowledge into soft labels, to train students who see only a fraction of those pairs, and are evaluated on the held-out remainder. We apply this protocol both to (i) small transformers on structured algorithmic tasks and (ii) fully connected networks on uncorrelated in- and output. While in (i) we exploit early stopping to obtain memorizing teachers, (ii) does not have a latent structure by design which always implies teacher memorization. Despite its simplicity, the controlled memorization-only setting (ii) has, to our knowledge, not been studied previously in the distillation literature.

For both cases we answer our original question *positively*: From training on the teacher's soft labels a student can indeed learn non-trivial information about held-out memorized data. A simple visual example for distillation in two dimensions is shown in Fig. 1. This has consequences both for the efficient transferability of memorized knowledge, as well as the leakage of private information. We summarize our specific contributions below[1]:

- We demonstrate for both structured but memorized datasets and purely random i.i.d. data that students trained on teacher's soft labels can consistently recover non-trivial – in some cases perfect – accuracy on data the teacher memorized but the student never saw.

- We show that this effect depends strongly on the temperature with which the soft labels are created from the teacher logits and can be interpreted as a regularizer that interpolates between fitting the teacher function and recovering only the ground-truth training labels.

- For random i.i.d. data, we show that in logistic regression, simple closed-form capacity and identifiability thresholds separate distinct leakage regimes, and that these thresholds extend to the multi-class case with similar qualitative behavior. For ReLU MLPs, the soft label memorizing and teacher-matching solutions are distinct; the student transitions from the former to the latter only once the teacher is identifiable, with a sudden jump in accuracy.

---

[1]Our results and the code to reproduce them are available at
https://github.com/SPOC-group/dataset-distillation-memorization.

## 2 RELATED WORK

**Distillation.** Soft labels have been central to *knowledge distillation* since its inception (Buciluǎ et al., 2006; Ba & Caruana, 2014; Hinton et al., 2015), and remain competitive across domains (Gou et al., 2021b; Xu et al., 2024). Their effectiveness has been linked to regularization effects (Müller et al., 2019; Yuan et al., 2020) and to encoding statistical structure aligned with the data distribution (Phuong & Lampert, 2019; Menon et al., 2021). Theoretical investigations of their effectiveness have considered simplified models such as deep linear networks or linear representations (Phuong & Lampert, 2019; Boix-Adsera, 2024; Dissanayake et al., 2025; Zhang et al., 2023). In particular, it is known that in the linear case distillation allows the student to efficiently match the teacher functionally Phuong & Lampert (2019) In a similar setting as ours, Saglietti & Zdeborova (2022) analyze regularization transfer from teacher to student, but take a teacher as a generating model itself rather than letting it memorize a fixed dataset.

In parallel, *dataset distillation* constructs small datasets that transfer capabilities for faster training (Wang et al., 2018; Yu et al., 2024), with similar effects observed even with arbitrary transfer sets (Yang et al., 2024; Nayak et al., 2021). These findings suggest that distillation success depends less on input realism than on whether the teacher function can be inferred from supervision (Cazenavette et al., 2023). In contrast, we analyze matched-capacity teachers and students on memorized data without input modification. While we do not modify the input distribution, our analysis shows that when the data is sufficient to functionally match the teacher, and softmax temperatures are high, the student can learn the teacher functionally rather than merely class labels. This contrasts with work on unlearning, where distillation is used to suppress specific capabilities robustly rather than retain unrelated capabilities (Lee et al., 2025). This difference highlights the importance of understanding when distillation preserves or erases information, as we do through different data regimes in toy examples.

**Memorization.** Zhang et al. (2017) famously showed that deep networks can fit easily random labels, demonstrating their large memorization capacity. We extend this observation by studying how such memorized information can be transferred via distillation with soft labels. This is relevant for modern large language models which do not memorize their training corpus, but simultaneously require factual recall (Chen et al., 2024). However, memorizing additional facts incurs a linear cost in model parameters (Lu et al., 2024). Bansal et al. (2022) distinguish example-level and heuristic memorization, where the latter relies on shortcuts or spurious correlations, which is known to hurt generalization (Bayat et al., 2025). In our random data setup, correlations in the dataset arise only from its finiteness, and our analysis in the large data and parameter limit rules out any spurious effects incurred by the finiteness.

**Privacy.** In privacy, the goal is often to create models that are only weakly dependent on individual training datapoints, preventing their recovery through queries or learning (Dwork et al., 2014). One specific setting focuses on hiding the labels of training data (Ghazi et al., 2021). This goal inherently contrasts with memorization, which requires retaining labels for single examples (Ma et al., 2024). While these objectives are rarely analyzed together, we consider a teacher that memorizes data and ask under what circumstances the label information of held-out teacher data can remain hidden from the student during soft label training. A practical example is given in (Cloud et al., 2025), where a student LLM is fine-tuned on random

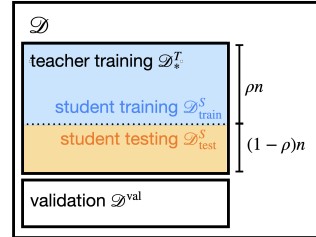

Figure 2: Data setting.

data sampled from a teacher that encodes a hidden trait. Training on unrelated memorized data causes the student to acquire the hidden trait, mirroring our observations in toy models. This illustrates that while soft labels can efficiently convey memorized teacher information, they can also create potential privacy risks, further motivating the analysis of our similar toy setting.

## 3 NOTATION AND EXPERIMENTAL SETTING

**Data.** We consider input-output pairs in a classification setting, where input coordinates are $\mathbf{x} \in \mathbb{R}^d$ and there are $c$ possible distribute labels $y$. The data is available either through the finite set $\mathcal{D}$ (Section 4) or a generating model from which we can sample i.i.d. (Section 5). See Figure 2. For

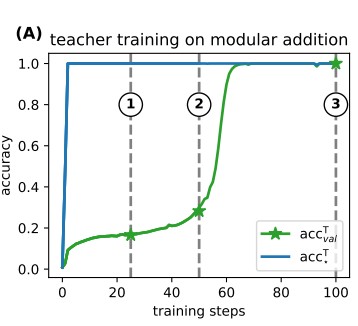
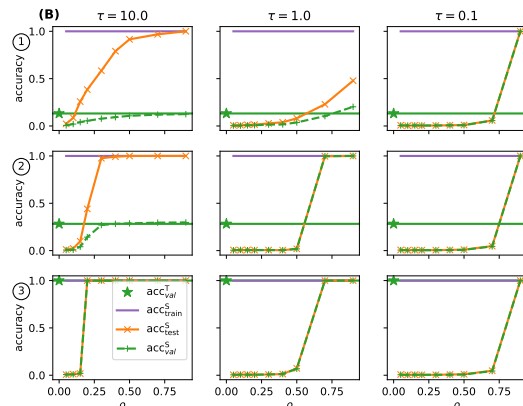

Figure 3: **Information leakage via soft labels for structured data in transformers.** **(A)** Loss curves for small transformers trained on the $30\%$ of the modular addition task with $p = 113$. The models ①, ② and ③ are stopped after different training times. **(B)** Students with a matching architecture trained on the respective teachers *(rows)* with different softmax temperatures $\tau$ *(columns)*. We show the students train and test error, and their accuracy on $\mathcal{D}_{\text{val}}$. For comparison, we show the teachers validation accuracy as a horizontal line, marked with a green star. Appendix C.1 describes architecture and training details. The same experiment is repeated for ReLU MLPs in Appendix C.3.

training the teacher we define the finite *teacher dataset* of $n$ such samples (elements) from $\mathcal{D}$ as $\mathcal{D}_\star^{\text{T}} = \left\{ (\mathbf{x}^\mu, y^\mu) \right\}_{\mu=1}^n$. To evaluate generalization of the teacher we consider $\mathcal{D}_{\text{val}}$, which is either $\mathcal{D} \setminus \mathcal{D}_\star^{\text{T}}$ or an independent sample. To train the student, the teacher dataset $\mathcal{D}_\star^{\text{T}}$ is randomly partitioned into two disjoint subsets: the student training set $\mathcal{D}_{\text{train}}^{\text{S}}$ and the student test set $\mathcal{D}_{\text{test}}^{\text{S}}$. We refer to $\rho = |\mathcal{D}_{\text{train}}^{\text{S}}|/n$ as the student's training data fraction.

**Models and Training.** All models we consider are parameterized functions $f_\theta : \mathbb{R}^d \to \mathbb{R}^c$ that map inputs $\mathbf{x}$ to class logits $\mathbf{z} \in \mathbb{R}^c$. Predictions are obtained by applying an argmax over the output logits. We use the cross entropy loss for supervised classification. For $\mathbf{y} \in \mathbb{R}^c$ being the one-hot encoded label vectors, for the teacher, the cross-entropy loss with temperature $\tau$ is

$$\mathcal{L}_{\text{CE}}(\{\mathbf{x}^\mu, \mathbf{y}^\mu\}_n) = -\sum_i \sum_{k=1}^c (\mathbf{y}^\mu)_k \log \left[ \sigma_{\tau=1} \left( f_\theta (\mathbf{x}^\mu) \right)_k \right] \quad ; \quad \sigma_\tau(\mathbf{z})_k \quad = \frac{\exp(z_k/\tau)}{\sum_{j=1}^c \exp(z_j/\tau)} \, .$$

To transfer the knowledge from a teacher $f^\star$ to a student $f_\theta$ we train them using the teacher's soft labels. This is achieved using cross-entropy loss, but instead of the ground truth one-hot vector $\mathbf{y}^\mu$ we use a given teacher network's soft labels $\hat{\mathbf{y}}^\mu = \sigma_\tau(f^\star(\mathbf{x}^\mu))$. We train using the Adam optimizer (Kingma & Ba, 2015) with full batches and default PyTorch settings (Paszke et al., 2019).

**Evaluation.** We report accuracies of the teacher and student: $\text{acc}_\star^{\text{T}}$ (teacher on $\mathcal{D}_\star^{\text{T}}$), $\text{acc}_{\text{train}}^{\text{S}}$ (student on $\mathcal{D}_{\text{train}}^{\text{S}}$), $\text{acc}_{\text{test}}^{\text{S}}$ (student on $\mathcal{D}_{\text{test}}^{\text{S}}$), $\text{acc}_{\text{val}}^{\text{T}}$ (teacher on $\mathcal{D}_{\text{val}}$), and $\text{acc}_{\text{val}}^{\text{S}}$ (student on $\mathcal{D}_{\text{val}}$). We always use the ground-truth labels to compute the accuracy for the student, and never the teacher predictions. When the teacher overfits the training data $\text{acc}_\star^{\text{T}} > \text{acc}_{\text{val}}^{\text{T}}$, we consider it to be memorizing. For $\mathcal{D}$ where the $c$ labels are sampled uniformly at random, independently of the input, both $\text{acc}_{\text{val}}^{\text{T}}$ and $\text{acc}_{\text{val}}^{\text{S}}$ reduce to random guessing at $1/c$, so teacher memorization requires $\text{acc}_\star^{\text{T}} > 1/c$.

## 4 LEAKING HELD-OUT MEMORIES WITH LATENT STRUCTURE

To complement our 2D toy setting from Fig. 1, we now study whether the leakage of memorized information through soft labels also occurs in more realistic architectures and structured data. Specifically, we use the modular addition task and a single layer transformer following Nanda et al. (2023). This setting training exhibits two phases: Even though the teacher quickly learns to fit the training set perfectly, generalization to the task is delayed. This allows us to isolate two different settings:

Teachers that memorize their training set without discovering structure, and those that generalize. From this, we examine how student learning varies based on what the teacher has learned and how this impacts the soft labels leakage of memorized information.

**Memorization and generalization in modular addition.** The modular addition task requires adding two integers $a, b \in [0, p]$ modulo $p$. We consider the case where this task is available as a dataset of tuples with one-hot encoded tokens $x = (a, b, p) \in \{0, 1\}^{3p}$ with the label $y \in [0, p - 1]$. For our experiments we consider only the case where $p = 113$, so that the number of possible unique samples is $|\mathcal{D}| = 113^2 = 12,769$. We train the teacher on 30% of this data, the set $\mathcal{D}_\star^{\mathrm{T}}$ ($n = 3,830$). The other part of the original dataset $\mathcal{D}$ is kept aside for validation in $\mathcal{D}_{\mathrm{val}}$. When training a student we split $\mathcal{D}_\star^{\mathrm{T}}$ into two disjoint sets $\mathcal{D}_{\mathrm{train}}^{\mathrm{S}}$ and $\mathcal{D}_{\mathrm{test}}^{\mathrm{S}}$, where $|\mathcal{D}_{\mathrm{train}}^{\mathrm{S}}| = \rho n$.
We train transformer architectures with a single layer (see Appendix C.1). To analyze teachers that have memorized the input data ($\mathrm{acc}_\star^{\mathrm{T}} > \mathrm{acc}_{\mathrm{val}}^{\mathrm{T}}$) to different degrees, we early stop the training, see Fig. 3(A). At checkpoint ① and ② there is memorization to different degrees, and at ③ the teacher generalizes. In the following, we use these teachers to train students via soft labels, and contrary to their early-stopped teachers we always train them until convergence. For a given experimental sweep, we select the maximum number of epochs such that the slowest converging setting run shows only small improvement, and apply this for all settings in this sweep. Since the phase transition in this specific setting where the student starts to be able to learn the data is at $\rho \approx 0.75$, a student with $\tau \to 0$ would obtain perfect generalization. We use different temperatures $\tau$ to generate the soft labels; results as shown in Fig. 3(B). Importantly, we observe that $\mathrm{acc}_{\mathrm{train}}^{\mathrm{S}}$ is always 100%, regardless of $\rho$ and $\tau$.

**Soft labels may leak memorized information in transformers.** In Fig. 3(B, first row, left), at $\tau = 10$ for teacher ①, we observe that for small $\rho$, i.e. small training sets $\mathcal{D}_{\mathrm{train}}^{\mathrm{S}}$, the student achieves higher $\mathrm{acc}_{\mathrm{test}}^{\mathrm{S}}$ (orange) than $\mathrm{acc}_{\mathrm{val}}^{\mathrm{S}}$ (dashed green). This means that indeed the soft labels are leaking some information on the training set $\mathcal{D}_\star^{\mathrm{T}}$, that accuracy on $\mathcal{D}_{\mathrm{test}}^{\mathrm{S}}$ is higher than for $\mathcal{D}_{\mathrm{val}}$. This indicates that the soft labels leak information specific to the teacher's training set $\mathcal{D}_\star^{\mathrm{T}}$ and allow the student to recover held-out memorized samples, while they do not improve performance on $\mathcal{D}_{\mathrm{val}}$ similarly strongly. As the fraction of seen teacher data $\rho$ grows, $\mathrm{acc}_{\mathrm{test}}^{\mathrm{S}}$ reaches 1.0, and $\mathrm{acc}_{\mathrm{val}}^{\mathrm{S}}$ approaches $\mathrm{acc}_{\mathrm{val}}^{\mathrm{T}}$: The teacher information is leaking. A similar but more abrupt transition occurs for teacher ② at the same $\tau = 10$ (middle row, left).
These results parallel our earlier observations from Fig. 1: For some $\mathcal{D}_{\mathrm{train}}^{\mathrm{S}}$ training on the teacher's soft labels leads to non-trivial accuracy on $\mathcal{D}_{\mathrm{test}}^{\mathrm{S}}$, which is importantly higher than that on $\mathcal{D}_{\mathrm{val}}$ (analogous to random guessing previously). Unlike the 2D case, however, here the student can *perfectly* generalize to the held-out $\mathcal{D}_{\mathrm{test}}^{\mathrm{S}}$. At the same time, despite $5\times$ longer training than the teacher, at $\tau = 10$, these students fail to generalize to $\mathcal{D}_{\mathrm{val}}$ when distilled from the non-generalizing teachers ① and ②. Instead they match $\mathrm{acc}_{\mathrm{val}}^{\mathrm{T}}$. This shows that while soft labels can leak memorized inputs, they can also prevent the student from learning latent structure that undertrained memorizing teachers have not discovered.

**Higher temperatures are more data efficient for fitting the teacher.** At lower temperatures $\tau$, where the soft labels resemble one-hot labels and contain less information about the teacher, the student can outperform the teacher and generalize to $\mathcal{D}_{\mathrm{val}}$. As shown in Fig.3(B, right column), at $\tau = 0.1$ student performance even becomes independent of the teacher. The student either fails to generalize due to insufficient data (e.g., at $\rho = 0.7$), or exhibits delayed generalization (learning curves Appendix C.2). Only for larger $\tau = 10$, learning from the generalizing teacher ③ requires less data with almost immediate generalization on $\mathcal{D}_{\mathrm{val}}$ and for the memorizing teachers ① & ② the students matches their function. This highlights that higher temperatures both improve data efficiency and convergence speed, and increase the leakage of teacher-specific memorized information.

## 5 LEAKAGE IN THE PURE MEMORIZATION SETTING

In the previous section, we used $\mathrm{acc}_{\mathrm{val}}^{\mathrm{T}}$ as a proxy for the amount of teacher memorization. However, a low $\mathrm{acc}_{\mathrm{val}}^{\mathrm{T}}$ does not rule out that the model internally captures some underlying structure, even if it was not predictive. To isolate memorization in a controlled setting and to characterize the leakage behavior theoretically, we now consider a data model where there is no structure in the data a priori – analogous to the introductory example from Fig. 1: The entries of the input $\mathbf{x}$ are sampled i.i.d.

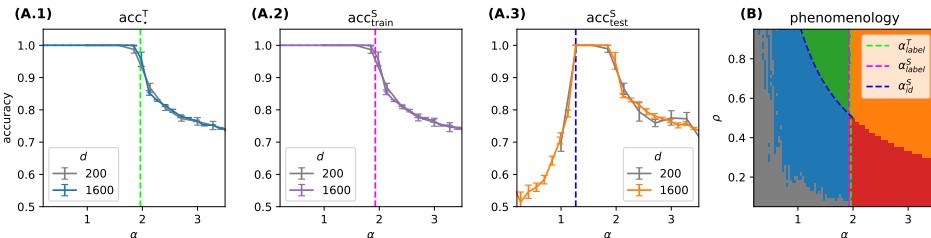

Figure 4: **Binary logistic regression. (A.1)** We show the training accuracy of the teacher on $\mathcal{D}_\star^\mathrm{T}$ for $\rho = 0.8$ and the students training **(A.2)** and testing accuracies **(A.3)** of the student on the two partitions of $\mathcal{D}_\star^\mathrm{T}$. While the teacher is trained via Adam on the logistic loss, the student solutions are obtained from the teacher logits on $\mathcal{D}_\mathrm{train}^\mathrm{S}$ using the pseudo-inverse. The thresholds $\alpha_\mathrm{label}^\mathrm{T}$, $\alpha_\mathrm{label}^\mathrm{S}(\rho)$ and $\alpha_\mathrm{id}^\mathrm{S}(\rho)$ are highlighted in green, pink and blue. **(B)** depicts the different regimes of teacher/student learning as a function of $\rho$ and the sample complexity $\alpha$. The dimension is fixed at $d = 1,600$ and $n$ is varied. We distinguish whether the student fits $\mathcal{D}_\mathrm{train}^\mathrm{S}$ with $\mathrm{acc}_\mathrm{train}^\mathrm{S} \geq 0.99$ (gray/blue/green) or not (red/orange). In the regime where it fits the $\mathcal{D}_\star^\mathrm{T}$, gray implies that the student learns only close to trivial accuracy ($\mathrm{acc}_\mathrm{test}^\mathrm{S} < 0.55$), blue that it is non-trivial ($\mathrm{acc}_\mathrm{test}^\mathrm{S} < 0.99$) and green is perfect ($\mathrm{acc}_\mathrm{test}^\mathrm{S} \geq 0.99$). We measure the MSE loss directly on the teacher logit (see Appendix D.1) to evaluate whether the student learned the teacher (orange) or not (red) – with a threshold set at 0.1.

from a Gaussian $x_i \sim \mathcal{N}(0,1)$ and the labels $y \in \{1,\ldots,c\}$ are sampled uniformly and i.i.d. from $c$ classes. The inputs and labels are independent by design, so any teacher needs to memorize the finite dataset $\mathcal{D}_\star^\mathrm{T}$, failing to generalize to $\mathcal{D}_\mathrm{val}$.

In the following, we analyze logistic regression, where we can derive closed-form thresholds for the recovery of $\mathcal{D}_\star^\mathrm{T}$ in the high-dimensional limit. We consider its multi-class version and show how the same threshold scales in $c$. To estimate the impact of more complex non-linear teachers we analyze leakage in one hidden layer ReLU MLPs. In Appendix B, we show that a teacher GPT-2 model (Radford et al., 2019) fine tuned on a dataset of randomly associated sequences of tokens and classes can also exhibit non-trivial test accuracy $\mathrm{acc}_\mathrm{test}^\mathrm{S}$ on held-out sequences.

## 5.1 MULTINOMIAL LOGISTIC REGRESSION

For multinomial logistic regression we consider linear models $f_\mathbf{W}(\mathbf{x}) = \mathbf{W} \cdot \mathbf{x}$ with $\mathbf{W} \in \mathbb{R}^{c \times d}$ that are trained via cross-entropy, known as multinomial logistic regression or softmax regression. In distillation this limits us to a setting where teacher and student architecture match.

**Formal analysis: leakage in logistic regression.** We first consider the case of only two classes, logistic regression[2]. When we have direct access to the logit, the problem of recovering the teacher weights $\mathbf{W}$ under the square loss is equivalent to solving an (over- or under-parameterized) least squares problem, by means of the pseudo-inverse of the input matrix with the logits, i.e., $\widehat{\mathbf{W}} = \mathbf{X}^+ \mathbf{z}$ where $\mathbf{X} \in \mathbb{R}^{n_\mathrm{train}^s \times d}$; $\mathbf{z} = f_\mathbf{W}(\mathbf{X}) \in \mathbb{R}^{n_\mathrm{train}^s}$ and $n_\mathrm{train}^s = |\mathcal{D}_\mathrm{train}^\mathrm{S}|$.

We consider different sample complexities $\alpha = n/d$. Fig. 4(A) shows the accuracy of the teacher on $\mathcal{D}_\star^\mathrm{T}$, and the train and test accuracies of the student on $\mathcal{D}_\mathrm{train}^\mathrm{S}$ and $\mathcal{D}_\mathrm{test}^\mathrm{S}$ as a function of $\alpha$ at fixed training set size $\rho = 0.8$. We observe that the $\mathrm{acc}_\star^\mathrm{T}$ and $\mathrm{acc}_\mathrm{train}^\mathrm{S}$ start decaying from 1.0 at a given $\alpha$. The test accuracy grows monotonically in $\alpha$ from the trivial random guessing accuracy up to perfect accuracy, and at some point it decreases again. The general phenomenology concentrates for large $d$ and $n$, as a function of $\rho$, resulting in three thresholds that can be defined in terms of $\alpha = n/d$:

$\alpha \leq \alpha_\mathrm{label}^\mathrm{T}$ – *teacher memorization capacity*: The teacher can fit all input-class pairs in $\mathcal{D}_\star^\mathrm{T}$. In the proportional limit when $d, n \to \infty$, Cover's Theorem (Cover, 1965) states that $\alpha_\mathrm{label}^\mathrm{T} \leq 2$ .

$a \geq \alpha_\mathrm{id}^\mathrm{S}(\rho)$ – *identifiability threshold*: The student functionally matches the teacher using the logits, measured through the mean squared error loss on the teacher logits, which occurs at $\alpha_\mathrm{id}^\mathrm{S} = 1/\rho$, as the input matrix $\mathbf{X}$ becomes invertible.

$\alpha \leq \alpha_\mathrm{label}^\mathrm{S}(\rho)$ – *student memorization capacity*: The student can fit all data from $\mathcal{D}_\mathrm{train}^\mathrm{S}$ via the input-logit pairs from the teacher.

---

[2]Here, we exceptionally consider $f : \mathbb{R}^d \to \mathbb{R}$ as this is the usual setup of the two class classification problem. The class is then determined by the sign of the single output logit $z$.

The capacity of the teacher $\alpha_{\text{label}}^{\text{T}}$ is independent of the distillation setting, stating how many randomly sampled inputs it can associate correctly to a randomly sampled but fixed label. In contrast, identifiability or functional matching of the teacher is possible from $\alpha_{\text{id}}^{\text{S}}(\rho)$, this threshold is identified for linear networks in Phuong & Lampert (2019). Finally, the student capacity $\alpha \leq \alpha_{\text{label}}^{\text{S}}(\rho)$ describes whether the student fits the teacher supervision from $\mathcal{D}_{train}^{S}$ correctly.

**In our experiments,** for finite sizes, we observe that the teacher memorization capacity $\alpha_{\text{label}}^{\text{T}}(d = 1600) \simeq 1.96$ is already close to the infinite $d$ limit of $\alpha = 2$. Beyond this threshold, the student cannot fit $\mathcal{D}_{\text{train}}^{S}$ perfectly anymore, as it is not memorized by the teacher and information is corrupted. However, when the teacher does memorize $\mathcal{D}_{\star}^{T}$ perfectly, the student obtains perfect accuracy on $\mathcal{D}_{\text{train}}^{S}$ through the logit training set. In this case, we observe that the logits contain a weak signal on the other held-out memorized data and allow the student to obtain $\text{acc}_{\text{test}}^{S} \geq 55\%$ for large enough $\alpha$ and $\rho$, as shown in Fig. 4(A.3); some information on the held-out data is leaking.
In terms of $\alpha$, $\text{acc}_{\text{test}}^{S}$ grows monotonically, e.g. for $\alpha_{\text{id}}^{S}(\rho = 0.8, d = 1600) \simeq 1.26$, where reaches $\text{acc}_{\text{test}}^{S} \geq 0.99$ – even though a fifth of the memorized data that was held-out. This means that the student can indeed recover the hidden memorized data by recovering the teacher weights $\mathbf{W}$.
Fig. 4(B) shows the different phases can be delineated as a function of $\alpha$ and $\rho$ for a finite fixed $d = 1600$: low/no leakage where $\text{acc}_{\text{test}}^{S} < 0.55$, weak leakage of information $\text{acc}_{\text{test}}^{S} \in (0.55, 0.99)$, full recovery of the held-out memorized data $\text{acc}_{\text{test}}^{S} \geq 0.99$ and failed teacher memorization beyond $\alpha_{\text{label}}^{\text{T}}$. We can separate the latter regime into two depending on $\rho$ and $\alpha$, whether the student is able to recover the (non-memorizing) teacher or not, depending on $\mathbf{X}$'s invertibility.

**The impact of temperature on memorization.** In practical distillation with more expressive networks one cannot simply invert but instead one minimizes the cross-entropy loss on soft labels via gradient methods. Creating soft labels from a students logits requires choosing a temperature $\tau$ in the softmax function equation 1. With $\tau \to 0$ one recovers the one hot encodings of the labels and thereby destroys any information that would have been embedded by the teacher. At the other limit, when $\tau \to \infty$, the soft labels become uniform and information about the labels and the teacher is destroyed.
For the case of multinomial regression with two classes Fig. 5(A) shows the relevant thresholds in terms of $\alpha = n/(dc)$ and on the temperature $\tau$ for a fixed $\rho = 0.8$ (for accuracies see Appendix D.2). Next to $\alpha_{\text{label}}^{\text{T}}$, $\alpha_{\text{id}}^{S}(\rho, \tau)$, and $\alpha_{\text{label}}^{S}(\rho, \tau)$, we introduce another threshold, $\alpha_{\text{label}}^{\text{S-shuffle}}(\rho, \tau)$, derived from a controlled experiment. For each input $\mathbf{x}$ with class $y$ in $\mathcal{D}_{\text{test}}^{S}$, we assign a soft label sampled from a different teacher input $\mathbf{x}'$ within the same class ($y = y'$). This procedure preserves the correct class identity – the highest soft label entry still corresponds to $y$ – but removes any teacher-specific information about $\mathbf{x}$. As a result, the student sees noisy supervision: It is class-consistent but the correlation between the rest of the soft label and input is broken. We then define $\alpha_{\text{label}}^{\text{S-shuffle}}(\rho, \tau)$ as the point at which this noise prevents the student from learning the class signal.
In Fig. 5(A), we observe that $\alpha_{\text{label}}^{\text{S-shuffle}}$ transitions from $\alpha_{\text{id}}^{S}$ to $\alpha_{\text{label}}^{\text{T}}$ as $\tau$ increases. This supports interpreting $\tau$ as a hyperparameter that shifts the training objective between fitting soft labels and teacher function (high $\tau$) and recovering class identity (low $\tau$).

**Multiple classes $c > 2$.** As the number of classes increases, the student has a $c$-sized soft label available per training sample, which can contain information about other samples. At the same time, the model size of both teacher and student scales with a factor of $c$. We observe empirically that the behavior for several classes is consistent with that for two classes: The student can learn non-trivial information about held-out memorized samples and achieve up tp $100\%$ accuracy from the soft labels. In Fig. 5(B) we observe the scaling behavior of the four relevant thresholds in terms of the number of classes $c$ for a *fixed* $\rho = 0.8$ and $\tau = 10$, leading to $\alpha_{\text{id}}^{S} \sim 1/c$, the scaling of $\alpha_{\text{label}}^{\text{T}} \sim \alpha_{\text{label}}^{S} \sim 1/\log c$ and $\alpha_{\text{label}}^{\text{S-shuffle}} \sim 1/\sqrt{c}$. Naturally, only the scaling of $\alpha_{\text{label}}^{\text{T}}$ is independent of $\rho$ and $\tau$. Specifically the scaling of $\alpha_{\text{label}}^{S}$ at $\tau \to 0$ should arrive at $\alpha_{\text{label}}^{\text{T}}$. Nonetheless, the order of the thresholds in $\alpha$ remains the same, retaining the original dependence. In Appendix D.2 we confirm this for varying temperatures and a fixed $c = 10$, where the phenomena are consistent with $c = 2$.

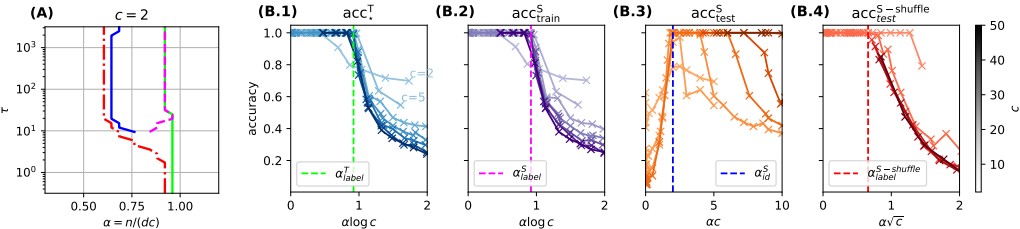

Figure 5: **Impact of (temperature $\tau$ — number of classes $c$). (A)** For the setting with $c = 2$ possible classes and $d = 1000$, we show the capacity and learning thresholds $\alpha_{\text{label}}^{\text{T}}$ (green), $\alpha_{\text{label}}^{\text{S}}$ (pink), $\alpha_{\text{id}}^{\text{S}}$ (blue) and $\alpha_{\text{label}}^{\text{S-shuffle}}$ (red) as a function of the softmax temperature $\tau$ and the sample complexity $\alpha$; accuracies are reported in Appendix D.2. **(B)** We take the number of classes as $c = \{2, 5, 10, 20, 30, 40, 50\}$, the larger $c$ the darker the color, and give train and test accuracies in varying scales $\alpha \cdot \{c, \sqrt{c}, \log c\}$. Here $\rho = 0.55$ is fixed and $d \in \{100, 1000\}$ is varied for computational efficiency depending on $\alpha$ and $\tau = 10$.

## 5.2 TWO MECHANISMS FOR LEAKING MEMORIZED INFORMATION IN ReLU MLPs

In this section, we show that ReLU MLPs already exhibit more complex behavior for the same random uncorrelated inputs and labels as before than the multinomial regression case. In Fig. 6 we consider a matched teacher and student, ReLU MLPs with a single hidden layer, with $c = 100$ classes. On the $x$-axis we vary the fraction $\rho$ of $\mathcal{D}_\star^{\text{T}}$ that is observed by the student. In Fig. 6(A), the $\text{acc}_{\text{train}}^{\text{S}}$ and $\text{acc}_{\text{test}}^{\text{S}}$ exhibit a similar phenomenon as before for the logistic regression in Fig. 4(D): While the student memorizes its own training set perfectly, the accuracy $\text{acc}_{\text{test}}^{\text{S}}$ on the held-out data is non-trivial and increases monotonically as more and more data from $\mathcal{D}_\star^{\text{T}}$ is available. However, at a higher sample complexity shown in panel (B) of the same figure, we observe two new phenomena: A first observation is the presence of a phase where $\text{acc}_{\text{test}}^{\text{S}}$ slowly drops while the teacher accuracy $\text{acc}_\star^{\text{T}}$ remains perfect. Meanwhile, $\text{acc}_{\text{test}}^{\text{S}}$ is *lower* than for the same $\rho$ at lower sample complexity $\alpha$. This is inconsistent with the previous observation, where larger $\mathcal{D}_\star^{\text{T}}$ helped functionally matching the teacher better and therefore led to higher accuracy. A second observation is the marked jump *after* the drop in $\text{acc}_{\text{train}}^{\text{S}}$, where both $\text{acc}_{\text{train}}^{\text{S}}$ and $\text{acc}_{\text{test}}^{\text{S}}$ immediately rise to $100\%$ accuracy.

**Memorization fails before teacher identification succeeds: $\alpha_{\text{label}}^{\text{S}} < \alpha_{\text{id}}^{\text{S}}$.** To understand these phenomena better, we turn to a more complete picture of the phase space in Fig. 7(A). Next to the regions already identified for the logistic regression in Fig. 4(B), we split the regions where a weak leakage is detected into two parts: The one where the student perfectly learns $\mathcal{D}_{\text{train}}^{\text{S}}$ and the one where the student does not memorize the training data. In Fig. 7(B.2) it is further visible that $\text{acc}_{\text{test}}^{\text{S}}$ decreases before it increases as a function of the sample complexity $\alpha$. To understand this behavior we observe $\text{acc}_{\text{train}}^{\text{S}}$ and $\text{acc}_{\text{test}}^{\text{S}}$ as functions of training time in Fig. 7(C.3). There, a sudden jump in train and test accuracy occurs as a function of student training time at around $t \sim 100$.

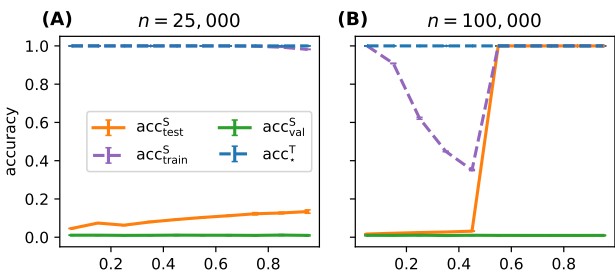

Figure 6: Both teacher and student are MLPs with a single hidden layer of size $p = 500$ and ReLU activations. The inputs are $d = 1000$ and $c = 100$. The teacher successfully memorizes a training set $\mathcal{D}_\star^{\text{T}}$ of size 25,000 **(A)** and 100,000 **(B)**. We track the accuracies on both via $\text{acc}_{\text{train}}^{\text{S}}$ and $\text{acc}_{\text{test}}^{\text{S}}$, with the standard error on the mean reported for 10 runs. In all cases shown here, some information is leaked statistically, allowing the student to surpass trivial performance on data not seen by the teacher ($\text{acc}_{\text{val}}^{\text{S}}$), in some cases reaching up to $100\%$ test accuracy.

While before the jump, the training and testing accuracy are at different levels (and already non-trivial for the student), they jointly jump to $100\%$ accuracy. In Appendix D.3 it is shown that this jump coincides with a drop in the CE loss on the teacher distribution and that just before the transition, $\text{acc}_{\text{train}}^{\text{S}}$ approaches that of a student trained on intra-class sampled soft labels. This suggests that for ReLU MLPs, there may be two distinct weight configurations: One where the student memorizes the

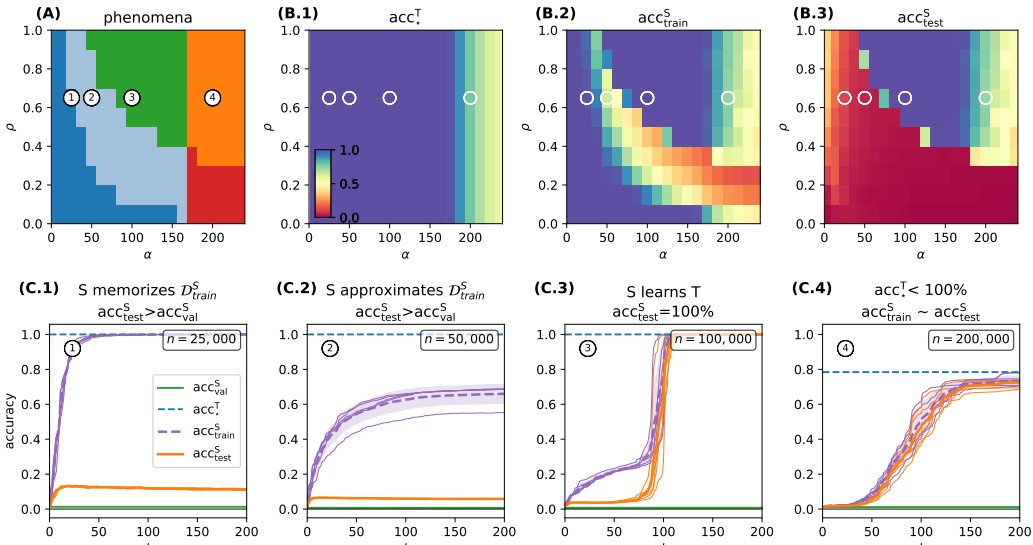

Figure 7: **Leakage in 1-hidden layer ReLU networks.** The teacher and student architectures match as single hidden layer ReLU networks with $p = 500$ for varying settings of sample complexity $\alpha = n/(dc)$ and student training fractions $\rho$. The number of samples $n$ is changed while $c = 100$ and $d = 1000$ and the temperature $\tau = 20$ are fixed. Each experiment is repeated 5 times and average accuracies are reported. **(A)** Different regimes distinguish the type of generalization the student achieves: (blue) weakly with memorization of $\mathcal{D}_{train}^{S}$; (light blue) weakly but without memorization of $\mathcal{D}_{train}^{S}$; (green) perfectly generalizing to held-out memorized data; (orange) the teacher cannot memorize $\mathcal{D}_{\star}^{T}$ but the student fits the teacher nonetheless; (red) the teacher cannot fit $\mathcal{D}_{\star}^{T}$ and the student does not discover the teacher either. **(B)** $\text{acc}_{\star}^{T}, \text{acc}_{train}^{S}$ and $\text{acc}_{test}^{S}$. **(C)** Accuracy as a function of training time $t$ for fixed $\rho = 0.65$ and different sample complexities $\alpha$ as marked with white circles in (A) and (B), varying $n$ and keeping $d = 1000$. For comparison, we show averages of $\text{acc}_{\star}^{T}$ and $\text{acc}_{val}^{S}$ at the end of training as horizontal lines.

soft labels, and another where it functionally matches the teacher. This distinction was not present for multinomial logistic regression.

**Memorizing the soft labels vs. generalizing on the teacher function.** These observations suggest that the student can learn two functionally different solutions that both leak information about held-out memorized data, but differently: One solution memorizes the teacher's soft labels representing $\mathcal{D}_{train}^{S}$, and another generalizing solution matches the teacher functionally. This extends the picture from the multinomial regression, in that not only weakly (and fully) learning the teacher function leads to non-trivial leakage on the held-out set, but also a solution that truly memorizes the soft labels can capture some additional structure on held-out data. Whether one or the other solution is learned depends non-trivially on the respective capacity thresholds, the algorithm, and the ratio between the teacher and student capacity. In Appendix D.4 we provide some additional ablations that explore how increasing the parameters via the hidden layer size $p$ and the relative capacity of teacher and student in an unmatched setting impact $\text{acc}_{test}^{S}$.

**Localizing the information in the soft labels.** In Appendix D.4 we test the effect of removing an input class $c_i$ from $\mathcal{D}_{train}^{S}$ and removing it from the soft labels by zeroing it out for all other classes $c_j \neq c_i$. We find that while removing the inputs can still lead to a non-trivial accuracy on $c_i$ in $\mathcal{D}_{test}^{S}$, removing the corresponding soft label entries is detrimental for test performance. Likewise, zeroing out the smallest $k$ values in every soft label negatively affects $\text{acc}_{test}^{S}$. This leads us to hypothesize that the common practice of using only the top-$k$ largest values may not allow for generalizing on the memorized information.

## 6 DISCUSSION AND CONCLUSION

In this work, we study how memorized data influences distillation. We analyze both structured data with teachers at varying levels of memorization or generalization as well as teachers that memorized data without a latent structure. We show that students can acquire information about memorized teacher data that was held out during their training from the teachers soft labels. By evaluating performances across teacher and student, we identify distinct regimes: In some, the student memorizes soft labels via statistical leakage; in others, it generalizes the teachers function. Our findings serve as a proof of concept to understand that memorized information can be transferred between models and extend prior results on linear students with binary labels by showing that this functional recovery behaviour persists across multiple classes and across nonlinear architectures. This broader empirical and theoretical picture indicates that the leakage mechanisms originally characterized for linear networks Phuong & Lampert (2019) apply more generally, which makes their implications for memorized data more concerning.

*Limitations.* Still, our analysis is restricted to synthetic datasets that are either structured or explicitly memorized and do not capture all aspects of natural data distributions. While this setup allows for precise control and analysis, it isolates the information contained in soft labels, which is the focus of our theoretical contribution, it limits the immediate application of our findings to real-world tasks even though such pipelines, for example privacy-constrained settings where the student has restricted data access and relies on a more permissive teacher, motivate the problem. Additionally, we focus on simple models such as logistic regression, single-layer ReLU MLPs and small transformers to enable theoretical clarity. Even though we show that leakage also occurs for one instance of a large language model with synthetic data, it remains unclear to what extent the identified leakage regimes and thresholds translate to deeper architectures. We did not analyze the influence of regularization or optimization dynamics, both of which can affect capacity and therefore leakage. Finally, we only consider memorization and soft labels, which excludes broader knowledge and dataset distillation settings where the teacher jointly learns generalizing structure and memorized data. Our results therefore isolate one mechanism through which soft labels convey information, rather than providing a full account of distillation in realistic settings.

*Future work.* On the theoretical side, our framework motivates a capacity analysis of multinomial logistic regression and single layer networks, to identify information-theoretic and algorithmic capacity thresholds for classification. On the practical side, it is important to understand how models represent both memorized and generalized content jointly, and in particular how this knowledge can be transferred for efficient dataset distillation, or whether it can be hidden for privacy reasons.

## ACKNOWLEDGMENTS

We thank Luca Biggio for insightful discussions and David Bau for pointing us to relevant related work. This work was supported by the Swiss National Science Foundation under grants SNSF SMArtNet (grant number 212049).

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

## A    VISUAL EXAMPLE FOR $d = 2$ AND $c = 20$

As another example that can be visualized analogously to Fig. 1, we show a random dataset again in two dimensions but with 20 classes in Fig. 8.

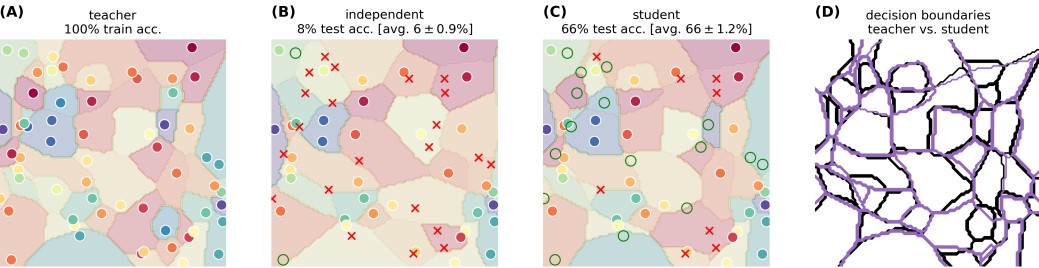

Figure 8: **Information leakage via soft labels for** $c = 20$. We examine fully connected networks with ReLU activations and $p = 300$ hidden neurons. A teacher network is trained on 2D input data $\mathcal{D}_\star^{\mathrm{T}}$ with i.i.d. random uniform labels drawn from 20 classes visualized in a spectrum of colors. **(A)** Visualizes $\mathcal{D}_\star^{\mathrm{T}}$ and teacher decision boundaries which achieve $100\%$ accuracy. Then, teacher data is partitioned into two disjoint sets $\mathcal{D}_{\mathrm{train}}^{\mathrm{S}}$ and $\mathcal{D}_{\mathrm{test}}^{\mathrm{S}}$ at $(60\%, 40\%)$ ratios. We examine 2 settings: Training student networks via cross-entropy **(B)** on the class information only, making the student independent from the teacher, and **(C)** on soft labels obtained from the teacher via softmax on the logits with temperature $\tau = 20$. While the independently trained model only achieves close to trivial accuracy of $\sim 6\%$, students that fit the teacher's soft labels achieve *non-trivial test accuracy* of $\sim 66\%$. Red and green indicate data from the test set, and whether it was classified wrongly or correctly. We show the average test accuracy and standard error on the mean over 5 runs. **(D)** The decision boundaries between teacher (black) and student (purple) correspond very well.

## B    DATASET DISTILLATION FOR FINETUNED GPT-2 CLASSIFIERS ON RANDOM SEQUENCES

In order to test whether the phenomena observed in Section 5 extends also to random sequence data, we examine a similar setting with a GPT-2 architecture (Radford et al., 2019). We consider sequences $x = \text{`}429\_3507\_345\text{'}$, where each sequence concatenates three random numbers sampled uniformly and i.i.d. between 1 and 1000, with a random class $y$ out of 1000 possible classes. In our setting $\mathcal{D}_\star^{\mathrm{T}}$ contains 6000 samples of such sequences and their classes. We equip the next-token prediction backbone GPT-2 with a linear classifier head. We use the standard tokenizer and train the teacher on $\mathcal{D}_\star^{\mathrm{T}}$ for 100 epochs using AdamW with a learning rate of $5 \times 10^{-4}$.

After successful training, when the teacher memorizes the sentences with $100\%$ accuracy, we extract the teacher's logits for its training data and create soft labels, with temperature $\tau = 20$. We train different the students for different fractions $\rho = \{0.2, 0.5, 0.8\}$ for 200 epochs, but otherwise use the same settings as for the teacher. After con-

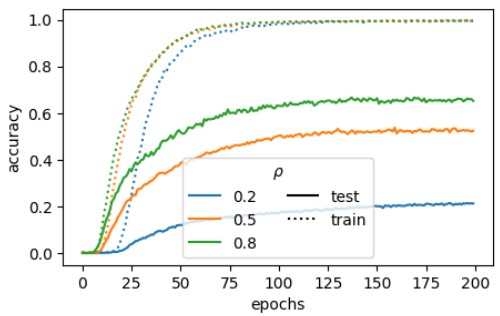

Figure 9: **Leakage in a large language model for synthetic data.** $\mathrm{acc}_{\mathrm{train}}^{\mathrm{S}}$ and $\mathrm{acc}_{\mathrm{test}}^{\mathrm{S}}$ for GPT-2 classifier students trained using different $\rho$ fractions of random sentences memorized by a teacher with the same architecture (both pre-trained). The size of the memorized training set is $n = 6000$ sentences made of three random numbers up to 1000, each with one of 1000 classes assigned randomly.

vergence all three students reach $\mathrm{acc}_{\mathrm{train}}^{\mathrm{S}} \simeq 99.5\%$ and $\mathrm{acc}_{\mathrm{test}}^{\mathrm{S}} = \{0.213, 0.524, 0.652\}$ (Fig. 9) – while the test accuracy of random guessing is approximately $\mathrm{acc}_{\mathrm{val}}^{\mathrm{S}} = 0.1\%$. For seeing only $80\%$ of the teachers data, the student achieves $> 60\%$ accuracy on the held-out data. This suggests that, similar to single-layer models, an over-parameterized language model may recover a non-trivial

fraction of the teacher's held-out memorized data. However, despite some exploration of different parameters, we did not yet observe a setting where the teacher function is exactly recovered as for the MLPs, i.e. where the student reaches $\text{acc}^{\text{S}}_{\text{test}} = 100\%$.

## C  SUPPLEMENTARY MATERIAL FOR MODULAR ADDITION

### C.1  IMPLEMENTATION DETAILS FOR THE TRANSFORMER

As an architecture we consider the single layer transformer from Nanda et al. (2023). It embeds the $p + 1$ tokens into 128 dimensions. There is a dot-product attention layer with 4 heads followed by a ReLU MLP with a single hidden layer of $4 \cdot 128$ dimensions. A readout layer maps its outputs to the $p$ classes.
By design the prediction is autoregressive, but in this case only the last predicted token is relevant and included in the training loss. It becomes a parameterized function $f : \mathbb{R}^{3(p+1)} \to \mathbb{R}^p$ where $c = p$ and $d = 3p$.
During training we use weight decay set to $1.0$ and use full batches, which recovers the setting in which grokking was observed (Nanda et al., 2023). We train using the Adam optimizer and a learning rate of $0.001$.

### C.2  GENERALIZATION SPEED

We examine how fast (in terms of training time) a student reaches perfect accuracy when trained on the soft labels from a perfectly generalizing teacher. In Fig. 10 we show exemplary learning curves for the results from Fig. 3 with teacher ③ in the main Section 4.

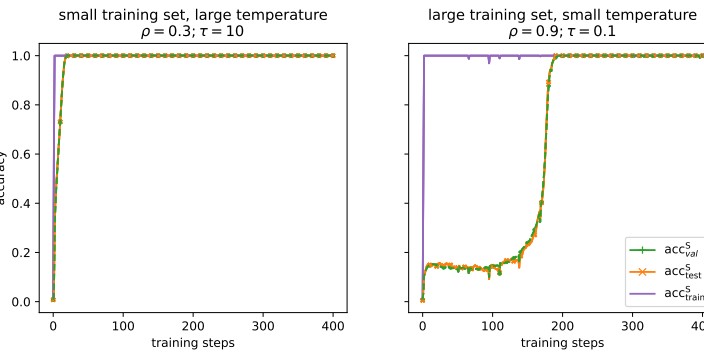

Figure 10: Comparison of the training, test and validation accuracy for transformer students (training and test data were seen by the teacher), for a single run from data used in Fig. 3(A). *(Left)* The student learning from a small training set ($\rho = 0.3$) with a high temperature ($\tau = 10$) not only memorizes its training data fast but generalizes on the held-out teacher train set and validation set after only few epochs. *(Right)* A student that sees a large training set at a smaller temperature however, exhibits grokking in a similar fashion as the original teacher (see Fig. 3(A) ).

### C.3  MODULAR ADDITION WITH MLPS

In addition to small transformers, we repeat the experiment from the main text Fig. 3 with 2-hidden layer ReLU MLPs with 200 hidden neurons each in Fig. 11.

We also stop the models training at three different points ①, ② and ③, which exhibit almost none, very weak and good generalization. In analogy to the transformer, at low temperatures the student's behavior becomes independent of the time at which the teacher training was stopped. Also, for higher temperatures and memorizing students, the $\text{acc}^{\text{S}}_{\text{test}}$ reaches higher values than $\text{acc}^{\text{S}}_{\text{val}}$, indicating that the soft labels only transfer information about the memorized labels but not the tasks structure and even prevent the students from generalizing.

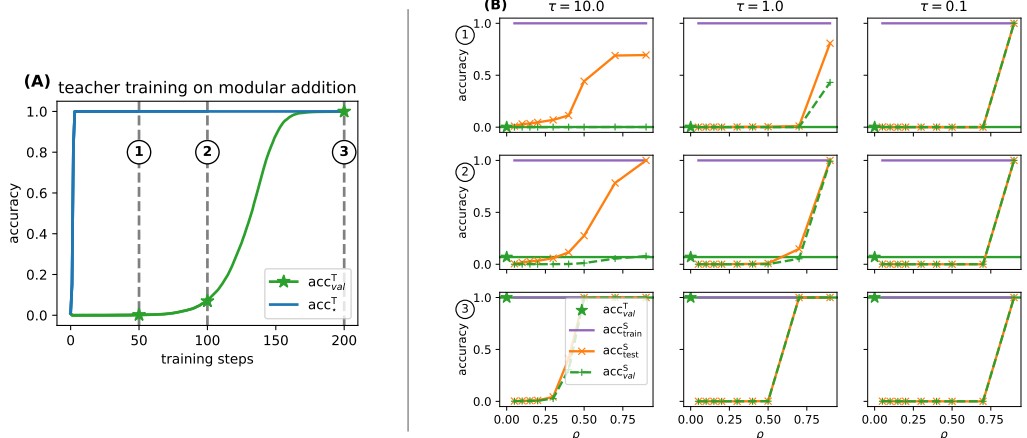

Figure 11: Same experiments as in Fig. 3, but with teacher and student architectures that are 2-hidden layer ReLU MLPs. **(A)** Teacher training that was stopped at different points in training, leading to the teachers ①, ② and ③. **(B)** Accuracies after training from the student for learning from the different teachers, at different $\rho$ and $\tau$.

## D SUPPLEMENTARY MATERIAL FOR RANDOM DATA

### D.1 LOGISTIC REGRESSION: TRAINING STUDENTS FROM THE LOGIT

We define the mean squared error between the teacher $f^T$ and the student $f^S$ on the dataset $\mathcal{D} = \{\mathbf{x}^\mu, y^\mu\}_{\mu=1}^n$ as:

$$\mathrm{mse}(f^T, f^S, \mathcal{D}) = \frac{1}{n} \sum_{\mu=1}^n \left( f^S(\mathbf{x}^\mu) - f^T(\mathbf{x}^\mu) \right)^2. \tag{1}$$

Let the matching accuracy of the student with respect to the teacher on the student test set be

$$\mathrm{acc}_{\mathrm{match\text{-}T}}^S = \frac{1}{n} \sum_{\mu=1}^n \mathbf{1} \left[ \arg\max_j f_j^S(\mathbf{x}^\mu) = \arg\max_j f_j^T(\mathbf{x}^\mu) \right], \tag{2}$$

where $f_j^T(\mathbf{x})$ and $f_j^S(\mathbf{x})$ denote the logits assigned to class $j$ by the teacher and student, respectively.

Fig. 12 shows the different values of $\mathrm{acc}_\star^T, \mathrm{acc}_{\mathrm{train}}^S, \mathrm{acc}_{\mathrm{test}}^S, \mathrm{acc}_{\mathrm{match\text{-}T}}^S, \mathrm{mse}(f^T, f^S, \mathcal{D}_{\mathrm{train}}^S)$ and $\mathrm{mse}(f^T, f^S, \mathcal{D}_{\mathrm{test}}^S)$ for $\rho$ and $\alpha = n/d$.

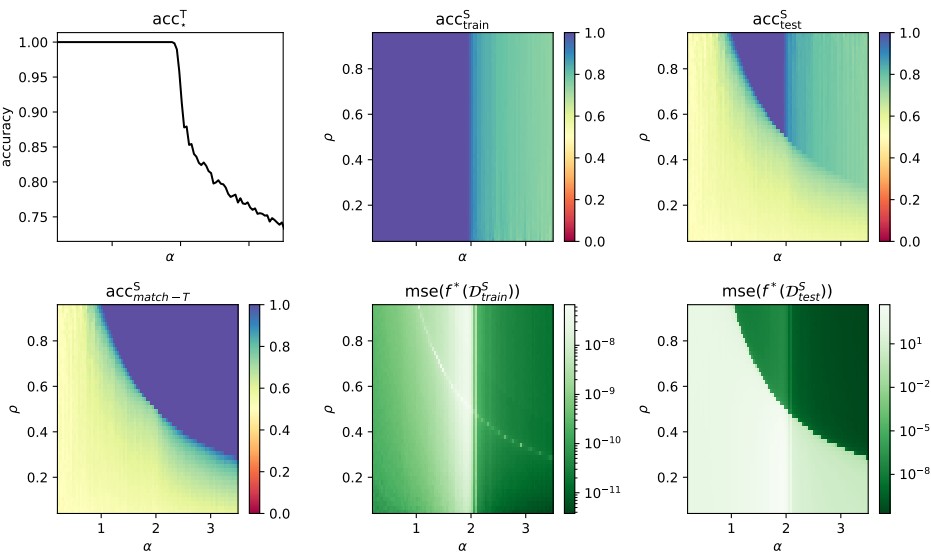

Figure 12: We train logistic regression teacher to fit $\mathcal{D}_\star^\mathrm{T}$, and recover a student via the teacher logit via a pseudo-inverse. We change the number of samples and keep $d = 1600$ while we change $\alpha = n/d$. Every point reports the mean of 5 experiments with different teacher, student and model initializations. For training the teacher we used Adam, learning rate $0.001$ and for $10,000$ steps.

## D.2 VARYING THE TEMPERATURE FOR MULTINOMIAL LOGISTIC REGRESSION

In Fig. 13 we show that the temperature influences whether or not held-out teacher data is leaked to the student for different $\rho$ and $\alpha$ in multinomial logistic regression with $c = 2$.

In addition, the original measurements that lead to the phase diagram for $c = 2$ in Fig. 5(A) are shown in Fig. 14. We show the same experiment for $c = 10$ in Fig. 15.

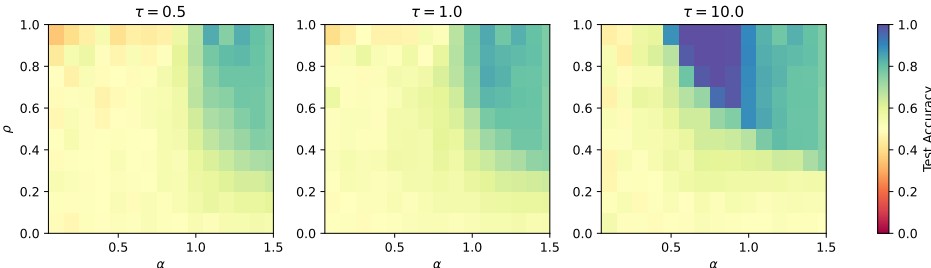

Figure 13: For multinomial regression with two classes and $\alpha = n/(dc)$, and $d = 1000$ we report the impact of different temperature in the softmax on $\mathrm{acc}_{\mathrm{test}}^\mathrm{S}$. Experiments are repeated 5 times. We train the teacher with Adam and learning rate $0.0001$ for $1000$ epochs and the student with learning rate $0.001$ for $5000$ steps.

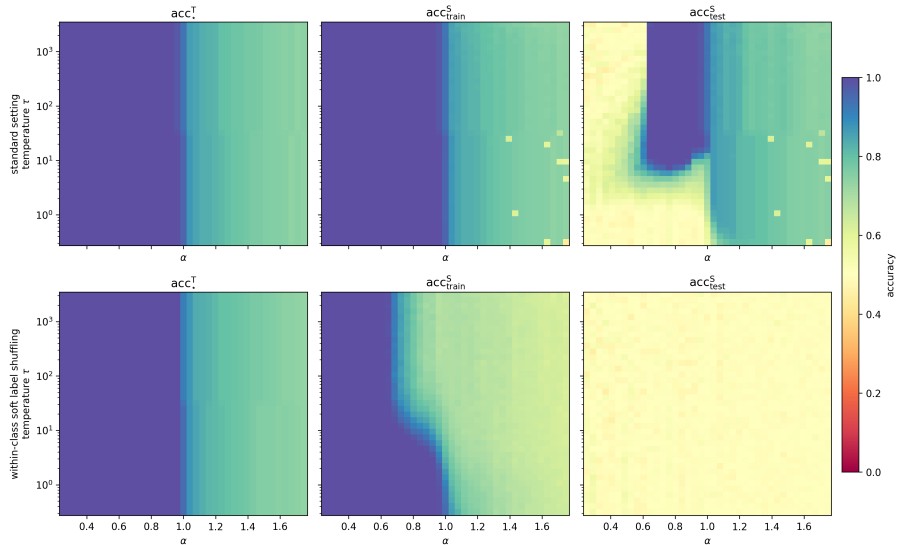

Figure 14: For multinomial regression with two classes $\rho = 0.8$ and $d = 1000$ we report accuracies for teacher and training data for two experimental settings. The top row is our standard setting and the bottom row re-shuffles the input-soft label assignment within the classes in $\mathcal{D}^{\text{S}}_{\text{train}}$.

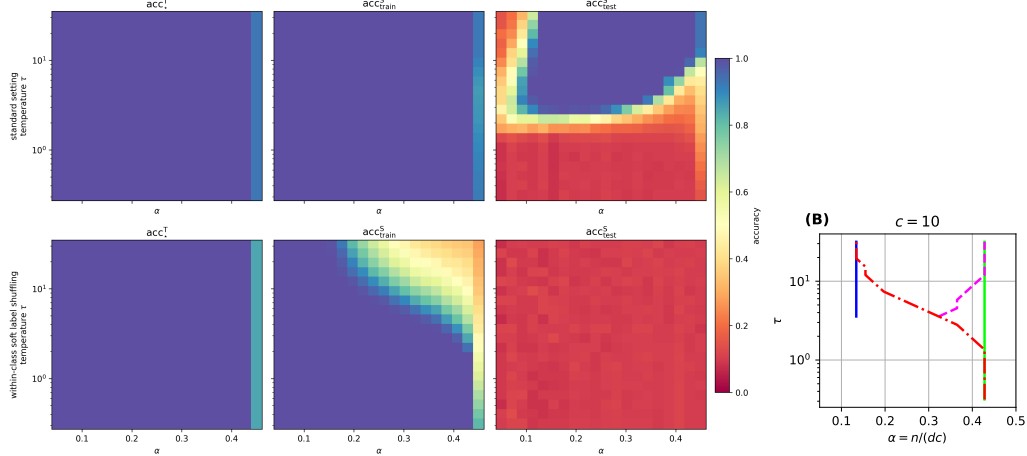

Figure 15: Same setting as Fig. 14, but with $c = 10$ classes instead of 2. The lines in (B) are the thresholds $\alpha^{\text{T}}_{\text{label}}$ (green), $\alpha^{\text{S}}_{\text{label}}$ (pink), $\alpha^{\text{S}}_{\text{id}}$ (blue) and $\alpha^{\text{S-shuffle}}_{\text{label}}$ (red) as a function of the softmax temperature $\tau$ and the sample complexity $\alpha$.

### D.3 SUPPLEMENTARY TRAINING INFORMATION

We supplement the accuracy curves from Fig. 16 for the single hidden layer ReLU MLP with the corresponding losses in Fig. 17, for different sample complexities $\alpha = n/(dc)$. It is visible, that around the same moment where the jump in accuracy occurs for Fig. 16(C.3), the loss also drops significantly.

In Fig. 16 we compare the learning curves from the students in Fig. 7(C) with those from the students which trains on the within-class shuffled soft labels. In the third panel it is visible, that before the student generalizes to the teacher function, it obtains the same accuracy as the shuffled student. Before this third panel, the accuracy is matching well, whereas for the forth panel the difference comes about quickly. This supports the intuition that the student first attempts to truly memorize, and then generalizes the teacher structure in the third panel.

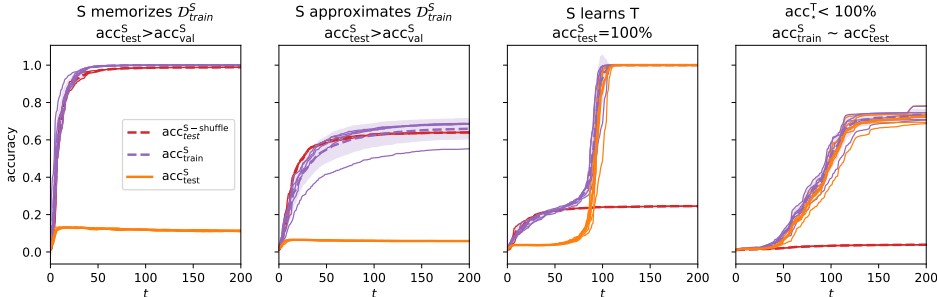

Figure 16: Accuracies for ReLU MLPs from Fig. 7(C), but this time compared with runs where the input data was shuffled within classes, in red. Again, the input dimension is $d = 1000$ with $c = 100$ classes and from left to right the teacher saw $n = 10^3 \times \{15, 50, 100, 200\}$ samples, of which the student was trained with a $\rho = 0.65$ fraction. There are 5 runs for the normal student, and 2 for the student that receives the altered teacher data for training.

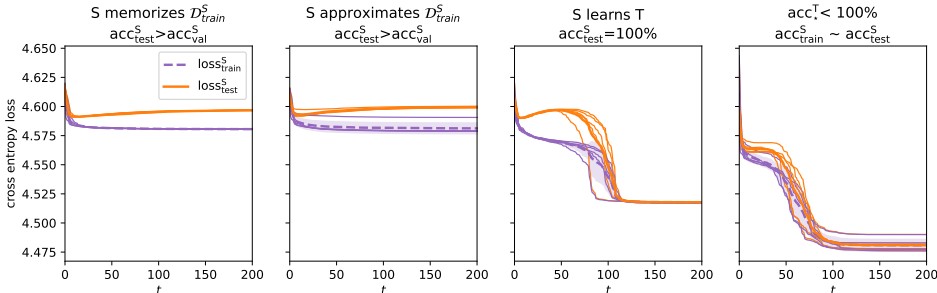

Figure 17: Cross entropy loss on $\mathcal{D}_{\text{train}}^{\text{S}}$ and $\mathcal{D}_{\text{test}}^{\text{S}}$ for the results from Fig. 16(C). The input dimension is $d = 1000$ with $c = 100$ classes and from left to right the teacher saw $n = 10^3 \times \{15, 50, 100, 200\}$ samples, of which the student was trained with a $\rho = 0.65$ fraction. We show the average (wide line) and single runs (thin lines) for 5 runs each.

## D.4 ABLATIONS FOR RELU MLPS

### D.4.1 VARYING THE SOFT LABEL CONTENT

In this section we conduct several ablations, to understand which information in the soft labels is crucial for obtaining a good $\text{acc}_{\text{test}}^{\text{S}}$. We conduct three experiments, where we:

- Remove small soft label entries: We zero out the entries of the smallest $k$ values of the $c$ soft labels for a given input. This leads to the rest of the vector not summing to one anymore, but the cross-entropy loss can still be computed.
- Remove a single class from the training data: We remove the class $c$ from the training data, and evaluate on the test data.
- Remove a single class from other classes soft label vectors: We specifically zero out the class $c$ value in the teacher soft labels for all classes $c' \neq c$.

In Fig. 18(A) we observe the effect of removing parts of the logits. Already removing a single entry is critical when the normal student would otherwise have learned the teacher. Removing more deteriorates performance in accuracy quickly, indicating that accessing the complete soft label is important to recover held-out memorized items.

In Fig. 18(B) we observe that removing the class $c$ from the soft labels is detrimental to the accuracy on that class in the test set ($\text{acc}_{\text{c}=1}^{\text{S}}$) but is maintained almost at a normal level for other classes. The average performance on the other hand is affected only little. In contrast, when we remove the class $c$ completely from the training set, but leave it intact in other classes' soft labels, the held-out sample

accuracy remains at a high level for class $c$, as well as the others. This further emphasizes that a lot of information is contained in the soft labels, and that especially the relational information to the class can help a lot.

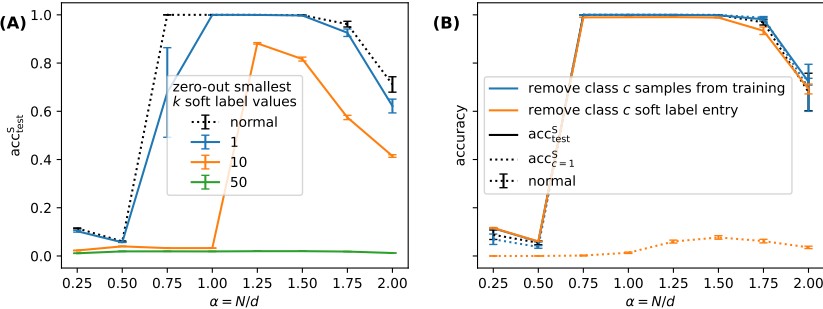

Figure 18: We consider ReLU MLPs with a single hidden layer where student and teacher have $p = 500$. We keep $d = 1000$, $c = 100$ and the fraction on which the student is trained is $\rho = 0.7$, and we use a temperature of $\tau = 20$. **(A)** We compare zeroing out the smallest $k \in \{1, 10, 50\}$ out of the 100 values from the soft label vector in the training data with a student trained on the unaltered data. **(B)** We also compare removing samples with class $c$ (here $c = 1$ w.l.o.g.) from the training data completely, removing the soft labels of all training data $\mathcal{D}^S_{\text{train}}$ whenever the true labels is not $c$. Every point is the average of 5 random datasets and initializations, with the standard error on the mean shown as bars.

### D.4.2 VARYING THE HIDDEN LAYER SIZE OF TEACHERS AND STUDENTS

In the following, we vary the sizes of the hidden layers in the teacher and student single layer ReLU MLPs, calling them $p^T$ and $p^S$ respectively. We first keep $\rho = 0.45$ fixed and keep the temperature $\tau = 20$. In Fig. 19 and 20, we vary $p^T$ and $p^S$ in isolation or together respectively.

In running our experiments for Fig. 19 we keep $d$ constant and vary $n$ and $p^T = p^S$, but instead of plotting the resulting accuracies over $\alpha = n/(dc)$ as in the main we plot them over $\alpha/p^S$. This incorporates the hidden layer size $p$ in the denominator that represents the number of parameters of the model, and indeed the curves fall together quite accurately.

In Fig. 19 we repeat the same experiment, but now keeping $p^T = 500$ and varying the student $p^S$. As we can see, this improves $\text{acc}^S_{\text{train}}$ for e.g. $\rho = 0.7$. This is expected as the student with more neurons has a higher capacity to learn the soft labels, which is the phase for the given $\rho$.

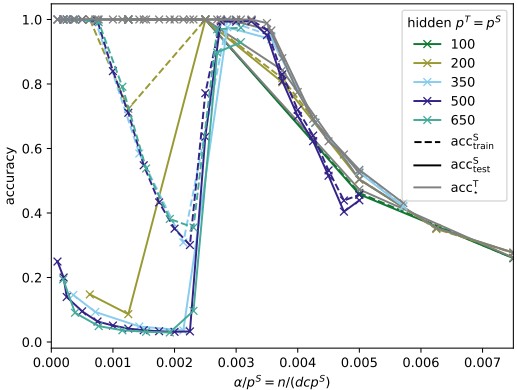

Figure 19: For ReLU MLPs we vary the teacher and students hidden layer sizes $p^T = p^S$ jointly. We keep $\rho = 0.45$, $c = 100$, $d = 1000$, and vary $n \in d \cdot \{12.5, 25, 50, 75, 100, 125, 150, 17, 200\}$, with hidden layer sizes as in the legend. Experiments here are repeated once.

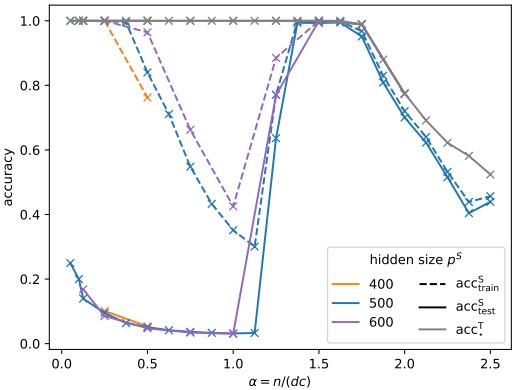

Figure 20: For ReLU MLPs we keep the teacher hidden layer sizes $p^T = 500$ and vary the studens hidden layer size $p^S$. We keep $\rho = 0.45$, $c = 100$, $d = 1000$, and vary $n$ with hidden layer sizes as in the legend. Experiments here are repeated once.

In Fig. 21 we zoom into the phase where the student is not matching the teacher but still finding a non-trivial $\mathrm{acc}^{\mathrm{S}}_{\mathrm{test}} > \mathrm{acc}^{\mathrm{S}}_{\mathrm{val}}$ on the held-out memorized teacher data. While we keep the teacher size fixed at $p^T = 500$ we observe that lowering the capacity of the student is optimal for all sizes of the dataset. On the other hand, for larger $p^S$ the accuracy decreases but not in a linear way - e.g. for $\rho = 0.3$ students with $p^S = 250$ are as good as students with $p^S = 1000$.

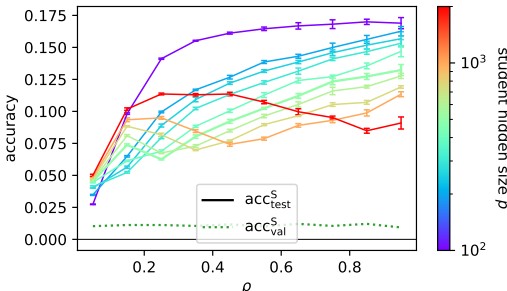

Figure 21: The teacher hidden layer size for the ReLU MLP is kept at $p^T = 500$, while we vary the data size fraction $\rho$ and the student hidden layer size $p^S \in \{100, 200, 250, 300, 400, 500, 600, 750, 1000, 2000\}$. We report the test accuracy and $\mathrm{acc}^{\mathrm{S}}_{\mathrm{val}}$ over the student with $p^S = 500$. Experiments are repeated 5 times and the standard error on the mean is reported.

### D.4.3 VARYING THE NUMBER OF CLASSES

Finally, we examine the effect of the number of classes in Fig. 22. When we scale the $x$-axis as $n/d$, the behavior we described in the main for the different classes remains similar.

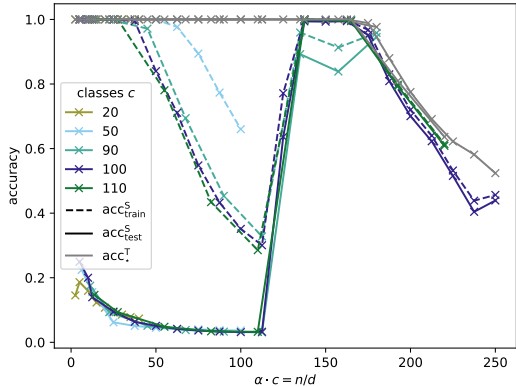

Figure 22: For ReLU MLPs with $p = 500$ for both teacher and student, we vary the number of classes $c \in \{20, 50, 90, 100, 110\}$ and show the student's accuracy on $\mathcal{D}^{\mathrm{S}}_{\mathrm{train}}, \mathcal{D}^{\mathrm{S}}_{\mathrm{test}}$ in a single run, and compare with the teacher accuracy.

## E  COMPUTATIONAL RESOURCES

All experiments can be run both on a CPU or GPU - for multinomial logistic regression a CPU may be faster than a GPU.

The most computational intensive were the phase diagrams Fig. 4a) with 13 compute days, and Fig. 6a) with roughly 10 compute days on a GPU. In both cases though, we ran the full pipeline for parallel experiments on a single machine with an NVIDIA RTX A5000 within roughly two days.

Since many of the experiments are run for different seeds to obtain error bars, running the experiments once is roughly a factor 5 faster than running all. Since experiments do not require large resources they can be parallelized easily.

