# OpenReview forum: "Dataset Distillation for Memorized Data: Soft Labels can Leak Held-Out Teacher Knowledge"
_ICLR.cc/2026/Conference — ICLR 2026 Poster_

### Official Review · Reviewer_Z77o · 2025-10-30

**Soundness:** 3
**Presentation:** 3
**Contribution:** 2
**Rating:** 4
**Confidence:** 2

**Summary:**

This paper studies an exotic problem: when a teacher trained to memorize a dataset and a student trained by distillation on a smaller training partition (compared to the teacher), the student generalizes better than a model trained on the same partition by standard cross entropy.

**Strengths:**

- The authors have done a meticulous and rigorous job in terms of clearly defining their hypothesis and constructing a specific set experiments which is used to study its validity.
- Although the experimental framework is constrained (the tasks studied and the networks used), the results presented seem convincing in terms of proving the authors' hypothesis

**Weaknesses:**

- Motivation: It is not clear what is the motivation of the presented experiments. Even if the paper's hypothesis is true what does that mean in practice and that are the applications? Why should the community pay attention to the results of this paper? While this is somewhat described in privacy paragraph of section 2, more clarity is needed to understand the paper's motivation.
- Impact of presented results: Can the paper's results be applied to real-world tasks, datasets and networks? Unfortunately, there's no evidence of that as also outlined in paper's limitations.
- What's the purpose of training on random data? Can this have some real-world implication or application?
- There's a very large number of experiments and cases analysed that makes the paper difficult to follow
- Better paper proof reading: e.g. line 235

**Questions:**

See weaknesses above.

---

> ### Author Response · Authors · 2025-11-23
>
> Dear reviewer Z77o,
>
> Thank you for the time taken in evaluating our manuscript.
>
> We address your concerns below:
>
> > Motivation and Impact
>
> As you mentioned, one practical setting is privacy-related training pipelines, where the student has restricted data access and relies on a teacher with broader access. Beyond this, the work is meant to contribute to the theoretical understanding of what information soft labels convey. Despite the widespread use of distillation, the mechanisms through which soft labels improve generalisation are still not well understood, and our controlled setting helps isolate one such mechanism with a focus on memorised information, giving insights into how a student can functionally recover the teacher and hence memories. We slightly modified the section on limitations to weigh this contribution against limitations in the real-world setting.
>
> > Flow
>
> More generally, our updated manuscript also contains some updates suggested by other reviewers. We hope that especially the dataset figure 2 helps in further clarifying our work
>
> We corrected the error you noticed on line 235 and gave the whole manuscript another re-read.
>
> We hope that our response clarifies your concerns, and hope that you also appreciate the additional modifications with respect to clarity and flow. If there are further questions or concerns you would like to discuss, feel free to get back to us.
>
> Best,
> The authors

---

### Official Review · Reviewer_hkBJ · 2025-10-31

**Soundness:** 4
**Presentation:** 3
**Contribution:** 3
**Rating:** 6
**Confidence:** 3

**Summary:**

This submission concerns information leakage in model distillation: what does the teacher pass to the student? In particular, when the teacher has memorized some labels from its training data, does this information pass through? The paper shows that the answer is yes. The main results are empirical results in controlled synthetic settings. There is some theoretical analysis as well.

Here's the main setup. We have some classification dataset $(X,Y)$ and train a teacher model $f(x)$ it. We split the data into $X=X_1 \cup X_2$ and $Y=Y_1\cup Y_2$. We train a student model on $(X,f(X))$. The experiments show how the student model can memorize the labels $Y_2$ without ever directly seeing them, i.e., the information comes solely through the teacher's labeling.

The teacher's predictions take the form of a probability distribution over classes and are controlled by a temperature parameter: in low-temperature limit the teacher makes hard choices and in the high-temperature limit the distribution is uniform. Clearly, in the high-temp limit, no information about $Y_2$ gets through to the student. On the other hand, if the teacher predicts the correct labels on its training data, then the low-temp limit corresponds to giving the student $(X_1,Y_1)$, which is independent of $Y_2$. But in between the soft labels seem to convey a lot of information!

The paper has lots of experiments on this phenomenon in different synthetic settings, with lots of interrelated results. It's not clear to me what the one big takeaway is, other than the fact that dramatic leakage can occur.

**Strengths:**

I enjoyed reading the submission. The experiments seem well-executed and are designed well: they do a good job probing the phenomenon.

I will likely study this paper further, and I expect many other people will find it interesting.

**Weaknesses:**

My main criticism of the paper is that the main takeaway, "soft labels can leak held-out teacher knowledge," appears to be known. Theorem 1 of Phuong and Lampert (2019), who the authors do cite as related work, shows that soft labels in a linear classification setting lead to the student recovering the teacher's weights exactly.

Now, this submission has a lot of results beyond the headline, so the work is still valuable. But the takeaways are less clear-cut. With so many experiments, the main part of this paper should probably be twice as long. The results are quite compressed and we jump around a lot. This paper would improve with additional work to craft the story.

**Questions:**

Is my understanding correct about the implications of Phuong & Lampert's Theorem 1?

Here is another natural setting for these experiments: the teacher trains model $f$ on $(X_T,Y_T)$ and the student trains on $(X_S, f(X_S))$, where $X_S$ and $X_T$ are independently drawn. Have you tried this? Would you expect very different behavior in this setting?

At points the experiments seem to switch away from cross-entropy, e.g., around line 295. Can you explain why we make that switch?

---

> ### Author Response · Authors · 2025-11-23
>
> Dear reviewer hkBJ,
>
> Thank you for taking the time to evaluate our work and to give some feedback. We answer your questions below.
>
> > Is my understanding correct about the implications of Phuong & Lampert's Theorem 1?
>
> It is indeed correct, that Phuong & Lampert show in Thm 1 that linear student networks with binary labels will learn the teacher weights and therefore recover the teacher functionally. They show that it only requires n ~ d samples. Beyond noting and discussing the implications of such behaviour for memorized data, in our setting we have several new findings and extensions. While the perceptron we discuss is clearly an instance of Phuong & Lampert, we show that this linear behaviour is also present when we train on several classes, in several architectures (including non-linear architectures). Therefore our setup broadens the implications of their very nice theory for linear networks. We hope that this understanding addresses your main criticism, i.e. that the main takeaway is known: We note that this behaviour is specially concerning for memorized data and that it transfers so robustly to a variety of settings, which is rather surprising and interesting. To highlight this further, we have improved the related work section and conclusion in the updated manuscript.
>
> > Here is another natural setting for these experiments:[training the student on fresh data labeled by the teacher]?
>
> We have conducted these experiments and we found that this does not seem to hinder training. This is conforming with the observation that arbitrary transfer sets sometimes also give good results in distillation [1], where the student probes the teacher at positions that are even far away from the
>
> > At points the experiments seem to switch away from cross-entropy, e.g., around line 295. Can you explain why we make that switch?
>
> This observation is correct: We briefly solve the perceptron with binary labels using matrix inversion since this gives the most efficient solution. However, everything else, including experiments on multiclass classification that follow for the perceptron, are keeping the cross-entropy.
>
> We hope that our response clarifies your concerns, and hope that you also appreciate the additional modifications with respect to clarity and flow. If there are further questions or concerns you would like to discuss, feel free to get back to us.
>
> Best,
> The authors
>
> [1] Effectiveness of Arbitrary Transfer Sets for Data-free Knowledge Distillation, Nayak et al 2020, https://arxiv.org/abs/2011.09113

---

> > ### Comment · Reviewer_hkBJ · 2025-11-24
> >
> > Thanks for your response. I retain my mostly positive impression of the paper.
> >
> > (If I were one of Phuong or Lampert, I might still grumble a bit: the centered & italicized questions on page 2 are already answered in some settings.)

---

### Official Review · Reviewer_h94i · 2025-11-02

**Soundness:** 3
**Presentation:** 2
**Contribution:** 4
**Rating:** 6
**Confidence:** 4

**Summary:**

This paper studies how memorized information can be transferred from a teacher to a student during dataset distillation through soft labels. The authors show that even when the teacher has not generalized, students trained on its soft labels can recover non-trivial information about held-out samples, both in structured and random data settings. Using controlled experiments with transformers, logistic regression, and ReLU MLPs, the paper identifies distinct regimes that separate different forms of information leakage, depending on sample complexity, temperature, and model capacity.

**Strengths:**

This paper tackles a very cool and relevant topic: whether dataset distillation, specifically using soft labels, can transfer memorized, held-out information from a teacher to a student. The work makes several very interesting observations on this front, and its primary strength lies in the careful experimental design used to isolate this phenomenon. The modular addition experiments, for instance, provide a compelling and clean demonstration of the difference in transfer outcomes between a generalizing teacher and a non-generalizing one. This, combined with the analyses on i.i.d. data using logistic regression and MLPs, creates a strong, controlled environment to probe the core questions of this work. The paper is exciting because it surfaces several intriguing phenomena—such as the clear importance of soft labels highlighted by the shuffling experiment and the complex behaviors hinted at in figures like 4A—which all point toward deeper mechanisms at play in distillation. The initial visualization in Figure 1 is also a very effective and intuitive way to ground the reader in the central concept.

**Weaknesses:**

The paper's primary weaknesses lie in the clarity of its presentation and the limited scope of the experiments.

* **Clarity and Focus:** The paper presents a wide array of experimental setups (modular addition, logistic regression, MLPs, varying temperature, classes, $\rho$, etc.) but struggles to synthesize them into a cohesive narrative. Many interesting results are condensed into dense paragraphs, making it difficult for the reader to dissect the core takeaways from each experiment.
* **Suggestions for Focus:** The paper's impact could be significantly improved by streamlining the main body. The authors might consider focusing on a few core experiments and analyzing them more deeply. For example, a clearer narrative could be built around:
    1.  The modular addition task, emphasizing the difference between generalizing and non-generalizing teachers.
    2.  The multinomial logistic regression with ReLU MLPs with a more thorough analysis of the role of $\rho$ (data fraction) and model scale (partially covered by the $\alpha$ experiment) in these settings.
    * Other analyses, such as the effect of the number of classes, seem less central and could be moved to the appendix to improve flow.
* **Unclear Notation and Definitions:** The theoretical discussion is difficult to follow due to ambiguous notation. The symbols $\alpha_{label}$ and $\alpha_{id}$ are introduced, but their practical meaning and the distinction between the "identifiability threshold" and "student memorization capacity" are not clearly explained.
    * To give an example, quoting: "*identifiability threshold: The student can identify the teacher using the logits, measured through the mean squared error loss on the teacher logits, which occurs at $\alpha_{id} = 1/\rho$, as the input matrix X becomes invertible.*"
    * The phrasing "the student can identify the teacher" is imprecise. It would be clearer to describe the empirical observation, such as "the student's predictions on the test set functionally match the teacher's predictions."
    * Similarly, naming $\alpha$ the "sample complexity" is not very intuitive; a term like "inverse overparameterization factor" might be more descriptive. A diagram illustrating the experimental setup and the relationship between these different $\alpha$ thresholds would be highly beneficial.
* **Figure Density:** Many figures, particularly Figure 4, are very dense and confusing. While Figure 4A seems to hint at a very interesting phenomenon, the key insight does not come through clearly from the visualization or the accompanying text.
* **Lack of Scale:** The authors acknowledge this as a limitation, but it is a significant one. All experiments are conducted on very small models. Without even medium-scale experiments, it is unknown whether these observed behaviors generalize to the deeper, more complex architectures where distillation is commonly applied.

**Questions:**

1.  Throughout the paper, the authors report the student's train, test, and validation accuracy. To be precise, are these accuracies calculated against the *original* ground-truth labels from the dataset, or against the teacher's (potentially incorrect) labels? The reviewer assumes the former, but it should be written explicitly.
2.  The paper uses several terms to define the data partitions ( $\mathcal{D}^{T}$ , $\mathcal{D}_{train}^{S}$, etc.. ). A small diagram visualizing the relationship and partitioning of these datasets, perhaps in the appendix, would greatly aid in understanding the experimental setup.
3.  The authors mention training models "until convergence" (e.g., in the context of Line 222). Could the authors please specify the exact convergence criteria used? For example, was it a fixed number of steps, a loss-based patience threshold, or another metric?
4.  In a similar vein to the user note on Line 215, the phrasing "the size of the complete data distribution" is ambiguous. Are the authors referring to the cardinality of the sample space, or something else?

---

> ### Author Response · Authors · 2025-11-23
>
> Dear reviewer h94i,
>
> Thank you for the time and care taken in preparing your review. We are answering your questions below:
>
> > (1) Throughout the paper, the authors report the student's train, test, and validation accuracy. To be precise, are these accuracies calculated against the original ground-truth labels from the dataset, or against the teacher's (potentially incorrect) labels? The reviewer assumes the former, but it should be written explicitly.
>
> Thanks for pointing this out. Your assumption is correct, so we updated the manuscript to include this information where we define the accuracies in line 195.
>
> > (2) The paper uses several terms to define the data partitions  … [A diagram illustrating the experimental setup and the relationship between these different thresholds would be highly beneficial.]
>
> Thanks for the suggestion, in the updated version of the paper we provide a small illustration in Fig. 2 that complements the texts and visually states the definitions of the data partition very quickly.
>
> > (3) The authors mention training models "until convergence" (e.g., in the context of Line 222). Could the authors please specify the exact convergence criteria used? For example, was it a fixed number of steps, a loss-based patience threshold, or another metric?
>
> In our experiments, we used a fixed number of gradient update steps for the various experiments. E.g. in Fig. 2 the number of steps is explicitly given. For the other settings, we observed how long the most challenging setting, i.e. the one with the largest number of parameters and largest memorization, took to converge, by observing a plateauing loss. We then set the maximum number of time steps accordingly. The precise values can be found in the accompanying code for each respective figure, defined through the number of epochs.
> We updated the manuscript to reflect this information around line 222.
>
> > (4) In a similar vein to the user note on Line 215, the phrasing "the size of the complete data distribution" is ambiguous. Are the authors referring to the cardinality of the sample space, or something else?
>
> For modular addition “a+b % p”with a and b restricted to a specific interval as mentioned in line 213, the maximal number of unique samples is “the size of the complete data distribution”. This is indeed ambiguous, we updated it to reflect that the size of the sample space is meant here.
>
>
> Weaknesses:
>
> > Unclear Notation and Definitions:
>
> You are correct, indeed we are only relying on empirical results to clarify alpha, therefore we revised any occurrence of the misleading phrase “The student can identify the teacher” to say “the student functionally matches the teacher on the test set”
> On the other hand, the term sample complexity (number of samples/parameter) is quite common in the literature, hence we will keep this term.
> Finally, it is indeed useful to understand these different thresholds in a more intuitive way, therefore we elaborated more carefully in the text. As they are introduced in Figure 3 and Figure 4 extensively reuses them, we moved Fig. 4B to the appendix, and instead only mention the scalings in the text.
>
> > Lack of Scale: The authors acknowledge this as a limitation, but it is a significant one. All experiments are conducted on very small models. Without even medium-scale experiments, it is unknown whether these observed behaviors generalize to the deeper, more complex architectures where distillation is commonly applied.
>
> In Appendix B of the current manuscript we include a setting, where we test the effect of memorization and distillation in a small language model, GPT2. We observe that the distillation leakage effect is also visible in the student for a teacher that was finetuned to memorize random random sequences of numbers. This gives an indication that the observed behaviour of leaking memorized information may generalize to more complex architectures, even though the model itself is still on the smaller side.
>
>
> We hope that our response clarifies how we expect our analysis to transfer to slightly more complex settings via the experiment on GPT2 and that you find the presentation of our work improved through more focused figures and an additional visualization of the setup. If there are further questions or concerns you would like to discuss, feel free to get back to us.
>
> Best,
> The authors

---

### Meta-Review · Area_Chair_aobm · 2026-01-04

**Summary:**

This paper only received three reviews. So I spent a lot of time thoroughly reading this paper. I share the same acknowledgement with other three reviewers regarding the contribution of the dense experimental results and the trigger of this topic of distillation memorization. Though the message this paper delivers is that distillation with soft labels will leak the held-out teacher knowledge, which is not new theoretically, as established in Theorem 1 of Phuong and Lampert (2019), cited by the authors, the paper presents more experiments on more complex cases to give stronger empirical evidence. Thus, I think this paper is slightly above the bar of ICLR, leading to a recommendation of week accept.
But I suggest the authors provide more a clear clarification about the theoretical contribution compared to the existing work in the final version

**Reviewer Concerns:**

Reviewer h94i's concern regarding the lack of a cohesive take-home message remains outstanding.

Reviewer hkBJ's concerns regarding the novelty of the main takeaway message is not well addressed.

Reviewer Z77o's concerns are solved.

**Reviewer Scores:**

Reviewer h94i and hkBJ already participated in the discussion and will maintain their socores.

Reviewer Z77o might raise his score to 4.

---

### Decision · Program_Chairs · 2026-01-26

Accept (Poster)